# Exploring Natural Products as Radioprotective Agents for Cancer Therapy: Mechanisms, Challenges, and Opportunities

**DOI:** 10.3390/cancers15143585

**Published:** 2023-07-12

**Authors:** Yi Zhang, Ying Huang, Zheng Li, Hanyou Wu, Bingwen Zou, Yong Xu

**Affiliations:** 1Division of Thoracic Oncology, Cancer Center, Department of Radiation Oncology, West China Hospital, Sichuan University, Chengdu 610041, China; kzy@live.com (Y.Z.); lizhenglys@126.com (Z.L.); 2College of Management, Sichuan Agricultural University, Chengdu 611130, China; 3Zhongshan Ophthalmic Center, State Key Laboratory of Ophthalmology, Guangdong Provincial Key Laboratory of Ophthalmology and Visual Science, Sun Yat-sen University, Guangzhou 510060, China; wuhy235@mail2.sysu.edu.cn

**Keywords:** radiation therapy, toxicity, radioprotector, medicinal plants, natural products

## Abstract

**Simple Summary:**

Natural products, especially plants and phytochemicals, show potential as radioprotective agents against radiation damage during cancer radiotherapy. They exhibit radioprotective effects through mechanisms such as free-radical scavenging, inhibition of inflammation, promotion of DNA damage repair, and inhibition of cell death pathways. Polyphenols, polysaccharides, alkaloids, and saponins from natural sources have radioprotective activity. However, clinical translation of these natural radioprotective agents is limited by issues such as low bioavailability, poor solubility, and high cost. Structural modification of natural compounds can improve their radioprotective efficacy and reduce toxic side effects.

**Abstract:**

Radiotherapy is an important cancer treatment. However, in addition to killing tumor cells, radiotherapy causes damage to the surrounding cells and is toxic to normal tissues. Therefore, an effective radioprotective agent that prevents the deleterious effects of ionizing radiation is required. Numerous synthetic substances have been shown to have clear radioprotective effects. However, most of these have not been translated for use in clinical applications due to their high toxicity and side effects. Many medicinal plants have been shown to exhibit various biological activities, including antioxidant, anti-inflammatory, and anticancer activities. In recent years, new agents obtained from natural products have been investigated by radioprotection researchers, due to their abundance of sources, high efficiency, and low toxicity. In this review, we summarize the mechanisms underlying the radioprotective effects of natural products, including ROS scavenging, promotion of DNA damage repair, anti-inflammatory effects, and the inhibition of cell death signaling pathways. In addition, we systematically review natural products with radioprotective properties, including polyphenols, polysaccharides, alkaloids, and saponins. Specifically, we discuss the polyphenols apigenin, genistein, epigallocatechin gallate, quercetin, resveratrol, and curcumin; the polysaccharides astragalus, schisandra, and Hohenbuehelia serotina; the saponins ginsenosides and acanthopanax senticosus; and the alkaloids matrine, ligustrazine, and β-carboline. However, further optimization through structural modification, improved extraction and purification methods, and clinical trials are needed before clinical translation. With a deeper understanding of the radioprotective mechanisms involved and the development of high-throughput screening methods, natural products could become promising novel radioprotective agents.

## 1. Introduction

Cancer poses a serious threat to human life and health, and in recent years, it has become a leading cause of death in humans. According to the American Cancer Society, there were 19.3 million new cancer cases worldwide in 2020, and 28.4 million new cases are expected worldwide by 2040 [1]. Radiotherapy, which has been used in cancer treatment for over one hundred years, is listed along with surgery and chemotherapy as one of the three main cancer treatment modalities [2]. More than half of all cancer patients receive radiotherapy at some point during their treatment, and radiotherapy is especially suitable for those with localized malignancies or solitary metastases [3]. Radiotherapy has the advantages of being safe and well tolerated, and its treatment effects are better than those of other therapies. Accumulated evidence on radiation has revealed that the damage caused by radiotherapy occurs through the following two principal mechanisms: direct and indirect (Figure 1) [2]. The direct effects are the consequence of the direct interaction of ionizing radiation with biological macromolecules, such as nucleic acids and proteins. The indirect effects result from radiation interactions, usually with water molecules, that cause the formation of free radicals. Specifically, organisms produce reactive oxygen species (ROS), such as hydroxyl radicals (•OH) and hydrogen peroxide (H_2_O_2_) [4], which can cause oxidative stress that leads to DNA damage and chromosomal aberrations after irradiation [5]. In fact, due to the large number of water molecules contained in cells, most radiotherapy-mediated DNA damage is caused by the indirect effects of accumulated ROS [6,7]. Further studies have revealed that ionizing radiation also transmits radiation signals from irradiated to unirradiated cells, a phenomenon known as the bystander effect. This effect regulates the levels of stress-related proteins and ROS in cells and ultimately leads to cell death [8]. Overall, ionizing radiation from radiotherapy exerts considerable effects in biological systems that can efficiently kill tumor cells. Modern radiotherapy techniques, such as three-dimensional conformal radiotherapy (3D-CRT), intensity-modulated radiotherapy (IMRT), and stereotactic body radiation therapy (SBRT), help to maximize the cytotoxic effects on tumors while minimizing the toxic effects on the surrounding normal tissue [2,9]. However, due to its low selectivity, radiotherapy also induces various degrees of damage to normal tissue. For example, irradiation from radiotherapy can cause toxicity in adjacent tissues that are sensitive to ionizing radiation, such as those in the hematopoietic, digestive, and immune systems [10]. In addition, the indistinctive boundaries between tumors and normal tissue increase the difficulty of preventing damage and toxicity to surrounding tissues.

In recent years, radioprotective agents, including sulfhydryl compounds, DNA-binding agents, cytokines, and hormones, have been used to decrease the toxicity to surrounding normal tissue [11]. However, most radioprotective agents are at the stage of animal experimentation, and further studies are needed to progress to clinical applications [11]. Currently, the only two drugs approved by the U.S. Food and Drug Administration (FDA) for reducing the adverse side effects of radiotherapy are amifostine and palifermin, neither of which is from a natural source. Amifostine is a sulfhydryl compound that reduces the renal toxicity and neutropenia caused by radiotherapy [12]. However, the apparent side effects, limited routes of administration, and limited protective effects on the central nervous system restrict its clinical implementation [13]. Ideal radioprotective agents are characterized by structural stability, high radiation protection, low adverse effects, convenient routes of administration, and highly selective activity that specifically protects normal tissues. Given the high toxicity and cost of synthetic chemical compounds, interest has developed in investigating natural products [14]. Natural products derived from plants, microorganisms, and marine sources have long been used for medicinal purposes and would be less toxic and less expensive radioprotective agents. Currently, natural products have been identified as efficacious in various diseases, including Parkinson’s disease, rheumatoid arthritis, inflammatory diseases, and cancer [15]. In addition, some natural products have shown different biological activities, such as antioxidant and anti-inflammatory activities and the promotion of DNA damage repair, which make them suitable for development as radioprotective agents [16,17].

These natural substances can come from a variety of sources, such as plants, fungi, or microorganisms, and they typically fall into several broad categories, including polyphenols, polysaccharides, alkaloids, and saponins. Polyphenols are a broad class of compounds that are characterized by the presence of multiple phenol units in their structure [16]. The most well-studied radioprotective compounds from natural products are apigenin, genistein, epigallocatechin gallate (EGCG), quercetin, resveratrol, and curcumin. Specifically, apigenin, found in parsley, celery, and chamomile tea, is known to possess antioxidant, anti-inflammatory, and anticarcinogenic properties [18,19]. Genistein, abundant in soy products, has been studied for its radioprotective effects, possibly due to its antioxidant activity and its capability to modify signal transduction pathways [20,21]. EGCG is a primary polyphenol in green tea that has been reported to protect against radiation-induced DNA damage [22,23]. Quercetin, which is widely distributed among plants, has a range of health benefits and can also protect cells against radiation [24,25]. Resveratrol, found in grapes and berries, has been shown to prevent radiation-induced damage due to its antioxidant and anti-inflammatory effects [26,27,28]. Curcumin, a component of turmeric, exhibits potent antioxidant, anti-inflammatory, and antitumor properties, thereby conferring radioprotection [29,30,31]. Polysaccharides are complex carbohydrates that have shown potential radioprotective effects. Astragalus, Schisandra, and Hoehenbuehelia serotina are notable examples. Astragalus is a traditional medicinal plant, and its polysaccharides boost the immune system activity and have antioxidant properties, offering protection against radiation-induced damage [32,33]. Schisandra polysaccharides have been shown to exhibit radioprotective effects through their antioxidant and immunomodulatory activities [34]. Hoehenbuehelia serotina, a species of fungus, produces polysaccharides that have shown potential radioprotective activity [35]. Saponins are a class of chemical compounds that are widely distributed in nature. Among them, ginsenosides from ginseng and acanthopanax senticosus have potential radioprotective effects. Ginsenosides, the major active components of ginseng, have been demonstrated to have radioprotective properties, possibly through their antioxidant, immunomodulatory, and anticancer effects [36]. Acanthopanax senticosus, also known as Siberian ginseng, is another source of saponins, which have been investigated for their radioprotective activity [37]. Alkaloids, a class of naturally occurring chemical compounds that contain mostly basic nitrogen atoms, also have potential radioprotective effects; matrine, ligustrazine, and β-carboline are example of these. Matrine, an alkaloid from the Sophora flavescens plant, has been shown to have radioprotective effects due to its anti-inflammatory and antioxidant properties [38]. Ligustrazine, an alkaloid from Ligusticum wallichii, has been found to mitigate radiation-induced lung injury [39]. The β-carboline alkaloids, which are found in several medicinal plants, are known for their wide range of pharmacological activities, including radioprotection [40]. In conclusion, these natural products from the polyphenol, polysaccharide, alkaloid, and saponin categories exhibit promising radioprotective properties. However, more detailed studies are needed to fully understand the mechanisms underlying these effects and to evaluate their safety and efficacy in clinical settings [41].

As we explore natural products for cancer therapy radioprotection, it is essential to pay homage to the scholarly groundwork and discern the limitations of earlier reviews in this field. Szejk et al., 2016 [42] provided a foundation by reviewing radioprotection by phytochemicals. Although primarily focused on plants, this review served as a gateway to subsequent research. Fischer et al., 2018 [43] summarized in some detail the natural products that may prevent radiation damage but did not elaborate on the mechanism of action. Mun et al., 2018 [44] explored the pharmacology of natural radioprotectors, providing an overview of several plant-derived compounds with radioprotective potential. However, their study did not offer a comprehensive review of various classes of compounds beyond plant-derived ones, and it lacked an in-depth discussion of the challenges. Adnan’s review in 2022 was instrumental in deepening the understanding of radioprotection by natural polyphenols [45]. Despite the valuable insights provided by these reviews, they had a narrow focus on specific classes of compounds or botanicals and did not address the translation from biochemical mechanisms to clinical applications. In contrast, this paper addresses these gaps and uniquely contributes to the literature by providing the first comprehensive systematic review of natural drugs as radioprotective agents, encompassing an exhaustive summary of various classes of compounds, including polyphenols, polysaccharides, alkaloids, and saponins. In this review, we briefly summarize the mechanisms of established radioprotectors, which may guide the development of new radioprotective agents. We also present the latest studies of various natural products that show radioprotective effects in radiotherapy. Finally, we discuss the opportunities and challenges associated with developing natural radioprotective agents. By integrating this wide array of natural compounds, bridging biochemical insights with practical applications, and highlighting the value of natural products as sources of developing new radioprotective agents, this paper offers a more holistic perspective that will pave the way for future research and development in the field of cancer therapy radioprotection.

## 2. Radioprotective Mechanisms of Natural Products

Radioprotective agents can mitigate the damage caused by radiation when administered either prophylactically (before exposure) or therapeutically (following exposure). Sulfur-containing compounds are mainly synthetic radioprotective drugs, but their application is limited by side effects. In recent years, natural products have received wide attention due to their low toxicity and multitarget mechanisms of action [44,46]. In this section, we mainly review the radioprotective mechanisms of natural products in cancer therapy (Figure 2).

### 2.1. Promoting DNA Damage Repair

DNA, being the primary repository of genetic information in the cell, is highly susceptible to damage induced by ionizing radiation, which can lead to structural alterations and functional impairments [47]. Some natural products have shown potential as radioprotective agents by promoting DNA damage repair [48]. For example, resveratrol, a polyphenolic compound found in grapes and berries, has been shown to promote the repair of radiation-induced DNA damage [49]. One of the mechanisms through which resveratrol exerts its protective effect is by promoting tyrosyl-tRNA synthetase (TyrRS) acetylation and inducing a cellular S-phase block via the downregulation of the deacetylase SIRT1 [50]. This modulation increases the efficiency of DNA repair through homologous recombination and ultimately reduces ionizing radiation-induced apoptosis. Cordycepin, derived from the fungus Cordyceps militaris, is another notable natural product that has shown efficacy in promoting DNA damage repair [51,52]. It functions as an agonist for nuclear factor-erythroid 2-related factor-2 (Nrf2) and is capable of preventing radiation ulcers [53]. The activation of Nrf2 by cordycepin leads to the upregulation of antioxidant defense genes. Additionally, cordycepin activates the AMP-activated protein kinase (AMPK) pathway, which plays a crucial role in DNA repair processes. Furthermore, curcumin, a polyphenolic compound found in turmeric, has been found to promote the repair of double-strand breaks (DSBs) in DNA, which are considered the most detrimental form of DNA damage [54,55]. Curcumin exerts its effects through the modulation of various signaling pathways, including the activation of DNA repair enzymes and the attenuation of oxidative stress, which contribute to the maintenance of genomic integrity [56]. Another example is quercetin, which is known for its antioxidative properties and is widely distributed in plants. Quercetin helps protect against radiation-induced DNA damage by enhancing the cellular antioxidant defense systems and promoting the repair of DNA strand breaks [57,58]. In conclusion, natural products such as resveratrol, cordycepin, curcumin, and quercetin offer promising avenues for promoting DNA damage repair following exposure to ionizing radiation.

### 2.2. ROS Scavenging

Water molecules are a significant source of free radicals in the body, contributing to approximately 80% of their production [59]. Ionizing radiation can cause the generation of ROS from water molecules, including •OH and H_2_O_2_. Among these, the hydroxyl radical is highly reactive and can instigate cellular lipid peroxidation [60]. ROS can inflict damage through the following three principal mechanisms: compromising cell membranes, inactivating serum antiproteases, and inducing genetic mutations due to DNA backbone breaks or base damage [61]. The body’s natural defense system employs several antioxidant enzymes for ROS scavenging, offering some level of protection against radiation [62]. In addition, natural products have shown a promising role in scavenging ROS and protecting against radiation-induced damage. Flavonoids have demonstrated notable ROS scavenging capabilities [63,64]. For instance, quercetin exhibits potent antioxidant properties and has been studied for its role in mitigating oxidative stress caused by ionizing radiation through ROS scavenging [65,66]. Phenylpropanoids, another class of natural compounds, also play a role in ROS scavenging. For example, caffeic acid, a type of phenylpropanoid, has been found to protect cells from radiation-induced damage by neutralizing ROS and reinforcing cellular antioxidant defenses [67]. Stilbenes such as resveratrol, primarily found in grapes and berries, have also been documented for their antioxidant properties. Resveratrol can scavenge ROS and modulate antioxidant enzyme activity; thus, reducing the oxidative damage induced by ionizing radiation [49]. Vitamin C is a well-known antioxidant that can neutralize ROS. It is a vital micronutrient for humans and can protect against radiation-induced oxidative stress by scavenging free radicals [68]. Additionally, active compounds from medicinal plants such as paeoniflorin, derived from the root of Paeonia lactiflora, have exhibited ROS scavenging capabilities. Paeoniflorin can alleviate oxidative stress by scavenging free radicals and modulating antioxidant enzyme activity [69]. *Inula racemosa*, a medicinal herb, is also noteworthy for its antioxidant properties. The active compounds in Inula racemosa have been found to scavenge ROS effectively and mitigate the cellular damage caused by ionizing radiation [70]. Ginsenosides, present in Panax ginseng, are saponins known for their antioxidant properties. They contribute to ROS scavenging and bolster the natural antioxidant defense mechanisms in cells; thus, protecting against radiation-induced oxidative damage [71]. In summary, various natural products, including flavonoids, phenylpropanoids, stilbenes, vitamin C, and active compounds from medicinal plants, play a significant role in ROS scavenging. By mitigating oxidative stress and reinforcing cellular defenses, these natural products hold promise as protective agents against the deleterious effects of ionizing radiation.

### 2.3. Anti-Inflammation

Radiotherapy can cause inflammation as a result of the indirect effects of ionizing radiation [72,73]. Various proinflammatory cytokines, such as interleukin-1 (IL-1), interleukin-6 (IL-6), tumor necrosis factor-α (TNF-α), and transforming growth factor-β (TGF-β), are produced in response to ionizing radiation exposure [74]. The release of inflammatory factors leads to increased vascular permeability and the activation of cyclooxygenase-2 (COX-2), resulting in apoptosis and fibrosis [75]. For example, TGF-β can induce lung and skin fibrosis under ionizing radiation [76]. Therefore, high TGF-β expression in lung cancer patients receiving radiotherapy is associated with an elevated risk of radiation pneumonitis [77]. Inhibition of TGF-β and other inflammatory factors reduces normal tissue damage, suggesting that reducing inflammation is a potential strategy for reducing normal tissue damage. For example, flaxseed (from *Linum usitatissimum*) has been verified to reduce the expression of lung injury biomarkers, such as BAX and p21, in mice, which is attributed to reduced lung inflammation [78]. Curcumin has been shown to downregulate the expression of proinflammatory cytokines such as TNF-α and IL-6, and inhibit the activation of nuclear factor kappa B (NF-κB), a vital regulator of the inflammatory response [75]. Genistein exhibits anti-inflammatory properties by inhibiting the secretion of proinflammatory cytokines and suppressing the activation of inflammatory pathways [79]. This can be particularly beneficial in reducing radiation-induced inflammation in tissues. Hesperidin, a flavonoid found in citrus fruits, has been shown to reduce inflammation by downregulating the expression of COX-2 and inhibiting the release of TNF-α and IL-6 [80,81]. Ferulic acid, a phenolic compound found in grains and certain vegetables, possesses anti-inflammatory properties [82]. It can inhibit the production of proinflammatory cytokines and reduce oxidative stress; thereby, mitigating radiation-induced inflammation [83]. Delphinidin, an anthocyanidin found in berries and colored fruits, has been studied for its anti-inflammatory properties [84]. It can inhibit the production of proinflammatory cytokines and modulate the signaling pathways involved in the inflammatory response [85]. Caffeine, commonly found in coffee and tea, has also been shown to have anti-inflammatory effects by inhibiting phosphodiesterase enzymes, reducing the production of inflammatory mediators [46]. In conclusion, natural products such as flaxseed, curcumin, genistein, hesperidin, ferulic acid, delphinidin, and caffeine, promise to manage and reduce radiation-induced inflammation [46].

### 2.4. Inhibition of Death Signaling Pathways

Inhibition of normal cell death signaling pathways is another way to protect against radiation-induced toxicity. For example, p53 is an important mediator of intracellular oxidative stress. After exposure to ionizing radiation, considerable cell death occurs in radiosensitive tissues due to the activation of the p53-mediated apoptotic signaling pathway [86,87]. Therefore, p53 inhibitors are considered protective in the context of radiation therapy [88]. Similarly, the mitogen-activated protein kinase (MAPK) pathways regulate fundamental cellular processes, and proteins in the MAPK family can modulate multiple signal transduction pathways that are stimulated by ionizing radiation [89]. Some natural products have been found to target and modulate these death-signaling pathways; thus, exhibiting radioprotective effects. Resveratrol is known to modulate the p53 signaling pathway. By inhibiting the activation of p53, resveratrol can reduce apoptosis in normal cells; thereby, protecting them from radiation-induced toxicity [90]. Curcumin has been demonstrated to inhibit the MAPK pathway [91]. Through this inhibition, curcumin can protect normal cells from radiation-induced apoptosis and enhance the radiosensitivity of tumor cells [92]. Epigallocatechin gallate (EGCG) is also known for its ability to modulate the death-signaling pathways. EGCG can inhibit the activation of p53 and exert anti-apoptotic effects on normal cells exposed to ionizing radiation, thereby mitigating radiation-induced toxicity [93]. MASM, a marine derivative, protects against radiotherapy-induced damage by modulating multiple MAPK pathways [38]. Additionally, Malyarenko et al. have shown that sulfated laminarin reduces the radiation resistance of melanoma cells in part by inhibiting MAPK signaling [94]. Ginsenosides, bioactive compounds from ginseng, can modulate the death-signaling pathways. For example, ginsenoside Rg3 can inhibit the radiation-induced activation of the MAPK pathways, which protects normal cells [68]. In summary, natural products such as resveratrol, curcumin, EGCG, MASM, sulfated laminarin, and ginsenosides can significantly inhibit the death-signaling pathways triggered by ionizing radiation. This makes them promising candidates for protection against radiation-induced toxicity. Further research is needed to unravel the underlying mechanisms of these compounds and evaluate their potential as adjuvants in radiation therapy.

## 3. The Radioprotective Effects of Polyphenols

Polyphenols, widely present as secondary plant metabolites, are phenolic compounds possessing multiple hydroxyl groups; based on their structural diversity, they can be classified into numerous subcategories, including but not limited to flavonoids and nonflavonoids, each with distinct physicochemical properties and biological activities (Figure 3) [95]. The biological properties of polyphenols, such as anticarcinogenic and anti-inflammatory activities, promote DNA repair, scavenge free radicals, and stimulate hematopoietic recovery and immune functions, which are the main mechanisms of radiation protection [96,97]. It has also been reported that some polyphenols can exert radiation-protective effects by regulating the expression of protein kinases and antioxidant enzymes [98].

### 3.1. Flavonoids

Flavonoids, which are low-molecular-weight polyphenolic compounds classified into six groups—isoflavonoids, flavanones, flavanols, flavonols, flavones, and anthocyanidins—share a common chemical structure characterized by two benzene rings [99]. Polyphenolic flavonoids are common constituents of plant-derived medicines and are present in a variety of plant species, such as parsley, onion, orange, chamomile, and tea. Flavonoids can modulate regulatory enzymes or transcription factors of inflammatory mediators and affect oxidative stress by interacting with DNA to enhance genomic stability [63].

Apigenin (4′,5,7-trihydroxyflavone), a natural plant flavonoid, is abundant in medicinal plants, fruits, and vegetables, including Chinese cabbage, bell peppers, guava, garlic, celery, parsley, and chamomile [19,100]. There is growing evidence demonstrating that apigenin has diverse biological activities, including antioxidant, anti-inflammatory, and antitumor activities [101]. For example, apigenin was shown to significantly reduce the number of micronuclei in γ-irradiated human lymphocytes in a dose-dependent manner [102]. Further study found that apigenin exerts radioprotective effects by scavenging the ROS caused by radiotherapy [103]. In addition, apigenin promotes the entry of Nrf2 into the nucleus, which activates the Nrf2 pathway and increases the expression of antioxidant genes [104,105]. Begum et al. [18] found that apigenin can reduce the lipid peroxidation index of lymphocytes caused by γ-radiation and increase the activity of superoxide dismutase (SOD) and catalase (CAT) in lymphocytes, indicating that apigenin has a good protective effect against radiation-induced oxidative stress. Recent studies have found that apigenin can inhibit the radiation-induced activation of NF-κB in gastrointestinal tissues; thus, exerting a radioprotective effect [106]. In these studies, safe and non-toxic concentrations were used. One study even escalated the administered dose to 200 mg/kg to amplify the plasma concentration of apigenin in response to higher radiation exposure, showing no observable toxicity [71]. However, despite the promising benefits of apigenin, it is characterized by poor systemic utilization due to its low solubility and short half-life. These properties could potentially limit its pharmacological effectiveness and clinical use.

Genistein (4′,5,7-trihydroxyisoflavone), a soy isoflavone, has been shown to play a role in radioprotection [107,108]. The radioprotective effects of genistein are mediated by several mechanisms, including free-radical scavenging, anti-inflammatory effects, and activation of DNA repair enzymes [109]. Genistein has been reported to decrease the hemorrhages, inflammation, and fibrosis caused by radiotherapy [110]. In animal studies, genistein reduced the radiation-induced intestinal damage in tumor-bearing mice [111]. In addition, genistein has been shown to increase the survival rate of mice that receive thoracic radiation [112]. A clinical trial indicated that genistein can reduce the adverse symptoms in the bladder and rectum in prostate cancer patients treated with radiation [113]. Notably, genistein has been found to have radiosensitizing and anticancer effects, as well as significant radioprotective effects [21,114,115,116,117]. However, the poor oral bioavailability and extensive metabolism of genistein restrict its clinical translation. Significantly, advancements in drug delivery systems have led to the development of a suspension of synthetic genistein nanoparticles, referred to as BIO 300. This novel formulation has been specifically designed to enhance genistein’s dissolution rate and oral absorption; thereby, optimizing its bioavailability [118]. Importantly, genistein also exhibits estrogenic activity, further diversifying its bioactivity profile and potential therapeutic applications. Despite these multifaceted attributes, genistein’s potential as a radioprotector has drawn particular attention, resulting in its investigational new drug (IND) approval for treating acute radiation syndrome. This IND approval denotes a pivotal step forward in recognizing the therapeutic potential of genistein and underscores the need for continued exploration in the field [119].

Epigallocatechin-gallate (EGCG) is the main component of green tea and has anti-inflammatory, antioxidant, anticancer, antiaging, antiarthritic, and neuroprotective effects [120,121]. EGCG exerts protection against radiation through a variety of mechanisms, including the scavenging free radicals, inhibiting lipid peroxidation, reducing DNA damage, and inhibiting ionizing radiation-induced ROS generation and apoptosis [22,23,122]. Specifically, it has been reported that EGCG can increase the expression of antioxidant enzymes such as SOD, glutamate cysteine ligase, and heme oxygenase-1 (HO-1) and is, therefore, recognized as an effective free-radical scavenger. In addition, EGCG can enhance DNA repair activity after ionizing radiation damage [123]. Pretreatment with EGCG was found to significantly decrease ionizing radiation-induced apoptosis and ROS production [124,125]. Furthermore, it was recently found that EGCG could significantly reduce ionizing radiation-induced damage to mice by modulating immunomodulatory activity [126]. EGCG can also reduce intestinal injury caused by total body irradiation in mice by modulating Nrf2 signaling and inhibiting ionizing radiation-induced apoptosis and ferroptosis [23]. However, EGCG is not only protective against radiation-induced damage at the animal or cellular level; it has also been demonstrated to reduce radiation-induced esophagitis [127,128], oral mucositis [129], dermatitis [130], and acute skin damage [131] in clinical trials. Overall, EGCG has good preventive and therapeutic effects on radiation injury, but the current level of evidence for clinical effects is insufficient, and the expansion of the clinical applications of EGCG is pending additional clinical trials.

Quercetin (3,3,4,5,7-pentahydroxyflavone) is a flavonoid found in a wide variety of plants, including berries, onions, shallots, apples, and tea [132]. Past studies have indicated that quercetin has anti-inflammatory, antioxidant, antibacterial, and anticancer activities [133,134]. Quercetin can reduce ionizing radiation-induced skin fibrosis by reducing the collagen accumulation and TGF-β expression in the skin [24]. Kale et al. [135] showed that quercetin significantly reduced the radiation-induced nerve cell degeneration and inflammatory infiltration. In addition, studies have shown that quercetin can attenuate radiation-induced bladder, lung, colon and kidney damage [25,136,137]. Thus far, the radioprotective effect of quercetin has been attributed to its anti-inflammatory [138] and antioxidant effects [139]. It has also been established that quercetin can act not only as a radioprotector but also as a chemosensitizer and radiosensitizer [140]. Quercetin has shown synergistic effects when combined with chemotherapeutic agents or radiotherapy [141].

Anthocyanins are a subgroup of flavonoids widely found in colored fruits and vegetables, such as blueberries, raspberries, purple kale, and purple potatoes [63], and they perform extremely diverse functions in plants [142,143,144]. Studies have shown that anthocyanins have free-radical scavenging, anticancer, and radioprotective activities. For example, Targhi et al. [145] showed that black mulberry anthocyanin reduced the levels of malondialdehyde (MDA) and SOD in rat liver and decreased the genotoxicity and cytotoxicity of γ-ray irradiation on rat bone marrow cells and liver, indicating a potential radioprotective effect. Liu et al. [146] found that blueberry anthocyanins significantly reduced the occurrence of radiation pneumonia through the expression of Bcl-2, Bax, and Caspase-3. However, high concentrations (greater than 100 mg/kg) limit the clinical application of anthocyanins.

Silibinin is a natural flavonoid that exerts anticancer activity in vitro and in vivo [147]. Several studies have confirmed the radioprotective potential of silibinin (oral treatment at 100 mg/kg/day) in radiation-induced late-phase pulmonary inflammation and fibrosis [148,149]. In addition, Prasad et al. [150] reported that silibinin prevented UV-induced thymine dimer formation and promoted DNA repair, and the initiation of apoptosis in damaged cells by increasing the level of the oncogene *p53*. Hesperidin is a naturally occurring flavanone glycoside that is mainly used as an antioxidant [151]. Studies have shown that hesperidin can inhibit the radiation-induced DNA damage in bone marrow cells; thus, exhibiting a strong radioprotective effect [152,153]. Furthermore, hesperidin showed antioxidant and anti-apoptotic activities in mouse testes that were damaged by ionizing radiation [154]. In addition, flavonoids such as caffeic acid [155,156], baicalein [157,158,159], vicenin [160,161,162], orientin [163,164], and chrysin [45,165] have also shown radioprotective effects. However, the relevant studies are limited, and further exploration is needed.

### 3.2. Non-Flavonoids

Resveratrol (3,4′,5-trihydroxy-stilbene) is a natural polyphenol that was discovered in 1939 and is derived from *Curcuma longa*, grapes, berries, peanuts, and other plant sources [166]. Studies on resveratrol suggest that this polyphenol has diverse bioactivities, including anti-inflammatory, antioxidative, antiapoptotic, and antitumor properties [166,167]. Carsten et al. [168] first reported the radioprotective effects of resveratrol, mainly through the induction of apoptosis, the scavenging of radiation-induced free radicals, inhibition of lipid peroxidation, and cell cycle arrest. In addition, Sebastià et al. [169] found that resveratrol could affect the transcription of antioxidant enzymes by regulating the level of protein kinase C to exert antioxidant properties and mitigate radiation-induced chromosome damage. Zhang et al. [170] found that resveratrol attenuated the radiation-induced intestinal damage in mice by activating Sirtuin1. Further research found that resveratrol has a protective effect against acute and subacute salivary gland damage by radiation [27]. However, resveratrol has the disadvantage of toxicity and cannot be absorbed when administered orally. In addition, the rapid metabolism of resveratrol in the intestine and liver limits its application [28]. Therefore, clinical trials on resveratrol remain limited [26].

Curcumin (1,7-bis-(4-hydroxy-3-methoxyphenyl)-hepta-1,6-diene-3,5-dione), a well-known bioactive polyphenolic compound, is the major active component of turmeric [171]. Curcumin has been used to treat cardiovascular disease, inflammation, and arthritis [172,173,174]. Currently, curcumin has been studied in combination with chemotherapeutic agents and radiotherapy in cancer treatment [92]. Moreover, curcumin has been demonstrated to exert radioprotective effects by scavenging ROS and inhibiting lipid peroxidation [175]. In addition, as a free-radical scavenger, curcumin can increase the activity of antioxidant enzymes, such as SOD, CAT, and glutathione peroxidase [176]. Another study demonstrated that curcumin can regulate the inflammatory factor levels, including that of IL-1, IL-6, interleukin-18 (IL-18), and TNF-α; thus, providing protection against radiation-induced damage [30]. Additionally, curcumin has been shown to protect the brain from carbon ion irradiation-induced irreversible cerebral injuries, including learning and memory defects [177]. Topical application of curcumin was found to reduce radiation-induced dermatitis and pain in a recent multisite, randomized, placebo-controlled, blinded study of 191 breast cancer patients [178]. Notably, curcumin has shown the dual action of radioprotection of normal cells while radiosensitizing tumor cells [29,31]. The major drawback of curcumin is its poor solubility in aqueous solutions. In addition, its poor absorption and rapid metabolism limit the use of free curcumin as a therapeutic agent; thus, drug delivery systems have been designed to reduce the impacts of these disadvantages. For example, radiation-treated mice were shown to have a higher survival rate when treated with curcumin-conjugated albumin-based nanoparticles [179].

Gallic acid (3,4,5-trihydroxybenzoic acid) is a phenolic compound found in various plants, such as green tea, grapes, and strawberries [180]. Nair et al. [181] found that gallic acid at physiologically relevant concentrations could significantly reduce radiation-induced DNA damage, prevent lipid peroxidation, and enhance the DNA repair process. Subsequently, Topsakal et al. [182] found that gallic acid significantly ameliorated the pancreas lesions induced by electromagnetic radiation. Materska et al. [183] found that phenolic glycosides isolated from peppers (*Capsicum annuum*) had good X-ray radioprotective activity in lymphocytes, effectively scavenging free radicals and reducing radiation-induced oxidative damage, and were not cytotoxic to human lymphocytes at all concentrations tested. Furthermore, studies have found that coniferyl aldehyde can reduce the radiation-induced genotoxicity and protect DNA from radiation damage [184].

## 4. The Radioprotective Effects of Polysaccharides

Polysaccharides are chains of more than 20 glycosyl groups formed by many monosaccharide molecules linked by glycosidic bonds [185]. Natural polysaccharides can be classified according to their origin as follows: plant polysaccharides, animal polysaccharides, and microbial polysaccharides. They are a class of biomolecules with immunomodulatory, antitumor, and anti-inflammatory effects [186]. The mechanism of the radioprotective effects of polysaccharides is not clear, but it is generally considered related to their antioxidant properties and their protection of the hematopoietic system, which enhances the efficacy of the immune system and induces the production of certain cytokines [187,188].

### 4.1. Plant Polysaccharides

Polysaccharides, naturally occurring in various plant organs such as flowers, roots, leaves, and fruits, have been found to exhibit a broad spectrum of biological activities in consumer organisms, including immunomodulatory, antioxidative, and antitumor activities. Plant polysaccharides have drawn increasing attention because of their biodegradability, sustainability, low processing costs, and low toxicity.

*Astragalus propinquus* is one of the most popular traditional medicinal herbs due to its diverse pharmacological activities, including antitumor, antiviral, antibacterial, and immune-promoting activities. *Astragalus* polysaccharides are polysaccharides from *Astragalus propinquus* isolated by extraction, separation, and purification. Liu et al. [32] investigated the protective effects of *Astragalus* polysaccharides against ^60^Co-γ radiation by using a Balb/c mouse model. It was shown that *Astragalus* polysaccharides significantly reduced the levels of alanine transaminase, aspartate transaminase, and lactose dehydrogenase in mice, and high doses of *Astragalus* polysaccharides also alleviated the ionizing radiation-induced liver and lung injury. In another study, Astragalus polysaccharides, at concentrations of several hundred µg/mL, were found to reduce the UVA-induced intracellular ROS production and mitochondrial membrane potential; thus, protecting the human keratinocyte cell line from UVA-induced damage [33]. However, importantly, these high concentrations may limit the potential for the clinical use of Astragalus polysaccharides.

Zhao et al. [34] found that *Schisandra* polysaccharide could prevent damage to the immune system by protecting immunoglobulins and inhibiting lymphocyte apoptosis, ultimately reducing the side effects caused by radiation therapy, which suggested that *Schisandra* polysaccharide may serve as a potential novel radioprotective agent. Acetylated mannan in aloe protected irradiated mice by regulating the immune system and stimulating the proliferation of hematopoietic cells [189]. Zhang et al. [190] found that *Rheum tanguticum* polysaccharide (RTP) could significantly reduce apoptosis and inflammatory factors by regulating Nrf2 and its downstream protein HO-1; thereby, significantly alleviating radiation-induced intestinal injury. Wang et al. [191] found that *Hohenbuehelia serotina* polysaccharides could effectively increase the activity of SOD and CAT in splenocytes and decrease the MDA content in splenocytes by mediating the endoplasmic reticulum apoptosis pathway, ultimately achieving radioprotective effects. Chen et al. [192] showed that *Sambucus nigra* substrate polysaccharide (SNAAP) could increase gluconeogenesis and glycogen synthesis in ^60^Co-γ-irradiated mice and restore radiation-induced disorders of glucose metabolism by regulating stress-activated protein kinase (JNK) pathways in the liver and pancreatic duodenal homologous box factor-1 (PDX-1) and glucose transporter protein-2 (GLUT-2) in the pancreas. In conclusion, while several polysaccharides, including Schisandra polysaccharide and Rheum tanguticum polysaccharide, have shown potential in radioprotection, the determination of safe and effective concentrations for clinical use is a critical factor that needs further exploration.

*Angelica sinensis* polysaccharide was found to exert radioprotective effects by increasing thymic and splenic indices and increasing the number of red and white blood cells in the peripheral blood [193]. Further studies showed that the radioprotective mechanism of *Angelica sinensis* polysaccharide was also related to increasing the antioxidant capacity of cells and reducing the DNA damage. Malyarenko et al. [94,194] found that polysaccharides from brown algae have radioprotective effects; for example, fucoidan is a sulfated polysaccharide found in brown algae. In addition to its antioxidant and antitumor activities, fucoidan can exert radioprotective effects in a dose-dependent manner [94,194]. In addition, plant polysaccharides, such as *aloe* polysaccharide [189], *Mesona blume* polysaccharide [195], cereal β-glucan [196], *Heliantnus tuberosus* L. polysaccharide, *Laminaria japonica* polysaccharide, Alga polysaccharide, *Haberlea rhodopensis* polysaccharide, purple sweet potato polysaccharide, soybean meal polysaccharide, pectin polysaccharide, *Panax ginseng* polysaccharide, and *Tinospora cordifolia* root polysaccharide, have demonstrated radioprotective effects.

### 4.2. Animal and Microbial Polysaccharides

Animal polysaccharides mainly include glycosaminoglycans and chitosan, which have antioxidant, anti-inflammatory, antibacterial, antiultraviolet, and other biological activities and may, thus, be used in drug development and in the biomedical field. Among the functional groups, the presence of uronic acid and sulfuric acid groups is closely related to the radioprotection of animal polysaccharides. For example, Li Na et al. [197] constructed a BALBb/c male mouse ^137^Cs-γ-ray radiation (4.0 Gy) injury model and treated the mice with siphon worm polysaccharide (SNP). SNP led to an increase in DNA and a decrease in the micronucleus rate in bone marrow cells, with a significant increase in antioxidant capacity and bone marrow damage protection. Moreover, a study found that chondroitin sulfate A derived from animals could increase the DNA content of mouse bone marrow after radiation therapy [198].

Polysaccharides derived from microorganisms are among the main types of polysaccharides, and several microbial polysaccharides have been found to have significant radioprotective effects. Lee et al. [199] found that *Lactobacillus* polysaccharides could improve the survival rate of zebrafish after radiation exposure, reduce the number of dead cells in the body, and decrease the production of ROS and nitric oxide in zebrafish; thus, demonstrating good radioprotective and antioxidant activities. Li et al. [35] found that neutral polysaccharides from *Hohenbuehelia serotina* could effectively increase the amount of bone marrow DNA, significantly inhibit the expression of Bax protein, and promote the expression of the Bcl-2 protein, which in turn inhibited the release of cytochrome c and promoted the expression of Caspase-3; thereby, blocking the mitochondrial apoptosis pathway in mouse spleen cells; this indicates that neutral polysaccharides have radioprotective effects.

## 5. The Radioprotective Effects of Alkaloids

Alkaloids are a class of organic compounds containing nitrogen with base-like properties, complex ring structures, and remarkable biological activities, including antitumor, anti-inflammatory, radioprotective, and other effects. Matrine is an alkaloid isolated from *Sophora flavescens* that has been demonstrated to attenuate the side effects of chemotherapy and radiotherapy by regulating and enhancing immune function in patients with cancer and synergizing with the therapeutic effects of chemotherapy and radiotherapy [200]. Ligustrazine (2,3,5,6-tetramethylpyrazine) is the major bioactive component of Rhizoma Chuanxiong. Zheng et al. [39] found that ligustrazine treatment at 20 mg/kg, 40 mg/kg, or 80 mg/kg concentrations attenuated the DNA damage and reduced apoptosis and inflammatory factor levels, indicating that ligustrazine is a good natural radioprotectant with significant radioprotective and antioxidant effects. β-carboline alkaloids are found in foods and plants and exhibit a variety of biological activities. Studies have shown that these alkaloids have antitumor, antiviral, and antimicrobial effects, and some experiments have also demonstrated their free-radical scavenging functions. β-carboline alkaloids were shown to be good scavengers of hydroxyl radicals in the Fenton system by Herraiz et al. [40]. Sverdlov et al. [201] found that β-carboline alkaloids can reduce oxygen-centered radicals and inhibit radiation-induced ethanol conversion; thus, functioning as antioxidants and radioprotectors.

## 6. The Radioprotective Effects of Saponin

Saponins are a class of glycosides based on triterpenes or spirosteroids that are not only found in vascular plants but also occur in small amounts in marine plants. Many herbs, such as ginseng and *Origanum*, contain saponins, which have been shown to have expectorant, antitussive, antibacterial, anticancer, and antioxidant activities, in addition to enhancing DNA repair. Ginsenosides are the main active components in ginseng, and they have strong antioxidant and radioprotective effects. Kim et al. [202] found that ginseng extract could reduce apoptosis and; thus, alleviate radiation-induced liver injury. Moreover, their study also found that 50 mg/kg and 100 mg/kg of ginseng extract showed consistently positive effects, but 25 mg/kg of ginseng extract did not. This means that the different effects occurred in a dose-dependent manner, which must be considered in clinical studies or use. Raghavendran et al. [36] found that galangal had a preventive effect on vomiting induced by X-ray radiation. In addition, ginseng extract protected rats from the damage caused by X-rays. *Acanthopanax senticosus* (Siberian ginseng) saponins at concentrations of 1.25 mg/100 mL–5 mg/100 mL protect neurons from free radical damage by scavenging oxygen free radicals, strengthening the cell membrane, and preventing oxidative stress [37]. Pokeweed saponin A protects soft tissues from radiation toxicity by inhibiting several proinflammatory cytokines and inflammatory mediators [203]. Gymnemagenin, a triterpene saponin, was shown to increase the CAT and SOD levels in the body and reduce the radiation-induced DNA damage in fish [204]. Kim et al. [205] found that timosaponin AIII inhibited medium-wave UV-induced COX-2 and matrix metalloproteinase-9 (MMP-9) transcription and protein expression in a dose-dependent manner, indicating its potential for development as a radioprotective agent.

## 7. The Radioprotective Effects of Other Natural Products

In addition to the polysaccharides, polyphenols, and saponins mentioned above, many other compounds in natural products have also demonstrated radioprotective effects (Table 1). For example, troxerutin (trihydroxyethylrutoside), a derivative of the natural flavonoid rutoside, is found in tea, coffee, cereals, and a variety of fruits and vegetables [206]. Troxerutin has been shown to selectively protect DNA in blood leukocytes and bone marrow cells (i.e., it does not protect tumor DNA). A dose-dependent radioprotective effect was also observed in mouse blood and bone marrow cells treated with troxerutin [207]. However, the specific mechanism of radioprotection by troxerutin remains unclear and may be associated with the activation of AKT and the inhibition of JNK, resulting in reduced ionizing radiation-induced phosphatase and tensin homolog (PTEN) activation [208]. Sesamol (3,4-methylenedioxyphenol) is a natural compound found in sesame. Some beneficial effects of sesamol have been reported, including antioxidation, antimutagenic, and antihepatotoxic properties [209]. In addition, sesamol is considered to be a potential radioprotective agent due to its strong ROS scavenging properties [210]. It was also found that sesamol significantly attenuated ionizing radiation-induced DNA damage in the hematopoietic system of mice and reduced the genotoxicity of bone marrow cells [211]. Lutein is a natural pigment widely found in vegetables, fruits, and certain flowers. Lutein has been reported to increase antioxidant enzyme activity; thereby, increasing its radiation-protective properties [212]. In addition, the radioprotective properties of several naturally occurring vitamins and dietary antioxidants have been tested [213]. For example, α-lipoic acid significantly increased mouse survival rates following lethal total body irradiation. Furthermore, vitamins A, C, E, and β-carotene have been found to increase radiation resistance [214]. Clearly, vitamins act as cofactors and regulate many physiological systems in addition to their radioprotective activities. These essential nutrients are required in small amounts by living organisms for normal function. Thus, they may harm rather than help at the high concentrations needed a priory for radiation protection purposes.

## 8. Limitations and Future Directions

Although preclinical studies have demonstrated the radioprotective effects of natural compounds, their clinical translation has been limited. Overall, there are many limitations in developing these natural compounds into drugs that could be used in the clinic, such as low bioavailability, water insolubility, excretion time from the body, and the high cost of extraction and isolation. First, one of the critical limitations hindering the clinical translation of natural products as radioprotective agents is the issue of effective concentrations. In many preclinical studies, natural compounds have demonstrated substantial radioprotective effects. However, these are often observed at concentrations of several hundred µg/mL, which may not be achievable or safe in a clinical setting [32,33]. For instance, certain natural compounds, such as polysaccharides and alkaloids, although potent in vitro, require high concentrations to exert their radioprotective effects [34,190,191,192]. These concentrations may far exceed what can be realistically and safely achieved in patients without causing toxicity or adverse side effects. It is crucial to recognize this disparity between in vitro effectiveness and in vivo feasibility, as it presents a significant challenge for the clinical application of these natural products. Future research should therefore focus on strategies to enhance the bioavailability of these natural radioprotectors and optimize their therapeutic index. Approaches such as nanoencapsulation, prodrug design, or coadministration with absorption enhancers may prove beneficial in this regard [122,158,179,219]. In addition, rigorous pharmacokinetic and toxicity assessments should be integral components of the development process for these natural radioprotective agents. Second, due to the time-consuming and labor-intensive screening methods for radioprotective natural products, it is important to establish high-throughput and highly specific screening methods for these natural products. In addition, the extraction and purification of natural products are currently very difficult, which is another barrier to clinical translation. However, with the gradual improvement in separation and purification technology, increasing numbers of natural active ingredients will be isolated and become easier to isolate, and the research and development of natural product radioprotective agents will have broad prospects. In addition to improving the extraction and purification efficiency of radioprotective natural products, the structural modification of existing natural products to obtain derivatives with higher radioprotective activity or fewer toxic side effects is another avenue for future research into radioprotective natural products. Specifically, it is well known that natural compounds are commonly found in the human diet with low intrinsic activity or are poorly absorbed in the intestine. Most natural compounds are rapidly metabolized under in vivo conditions so that only a minute percentage of free phytochemicals are retained in the bloodstream to the extent that they do not act effectively. To overcome these limitations, the molecular structure of the compounds can be modified to increase their water solubility. Rutin is a flavonoid that can be found in a wide range of plants. Its ability to inhibit long-living radicals has been shown, stressing its role as a radioprotector [224]. Despite the significant benefits of this natural compound, the use of rutin has been impaired due to its low water solubility [225]. A water-soluble derivative of the compound or a slightly modified compound was investigated to address this issue. A new form of rutin, αG-rutin, was synthesized by introducing a glycosyl group to rutin. Compared with rutin, its solubility in water is improved by a factor of 30,000 or more [226]. Applying another phytochemical component to the compound of focus and using it as a mixture could also increase its bioavailability [227]. In addition, a compound could be prepared for oral administration to increase its bioavailability [228]. For example, it has been found that the bioavailability of apigenin in rats can be increased by the oral administration of apigenin and friedelin [229].

## 9. Conclusions

The development of radioprotective agents is imperative in cancer radiotherapy. Although many synthetic drugs have shown effective protection against radiation, their practical application is limited due to their high toxicity. Several plants and phytochemicals have been shown to be effective in radioprotection during radiotherapy, as they exhibit both radioprotective and anticancer properties. Compared with traditional chemically synthesized drugs, drugs derived from natural sources have the advantages of high activity, high selectivity, and low toxic side effects, and have broad developmental prospects as radioprotective drugs. However, most of this research is only at the stage of in vitro or in vivo experimentation, and few products have been used in clinical applications. To develop these phytochemicals into clinically effective drugs, the biological characteristics of these natural products with radioprotective effects and the mechanisms of their radioprotection must be studied in depth. Furthermore, relevant in vivo trials should be accelerated to assess the toxicity and pharmacokinetics of the drugs in preparation for clinical trials. Moreover, most of the recent studies include single-component antiradiation drugs, and we hope that researchers build on what has been found and exploit the characteristics of each compound to screen for low toxicity, effective compound radiation protection agents. In conclusion, the use of plant-based radioprotectors as an adjunct to radiotherapy is effective and safe, and they may serve as potential candidates for radioprotection soon.

## Figures and Tables

**Figure 1 cancers-15-03585-f001:**
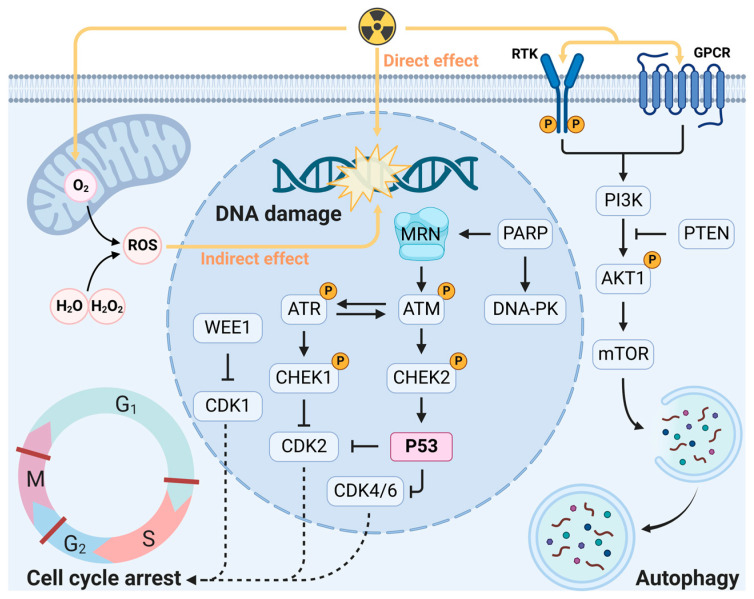
Ionizing radiation can directly or indirectly damage DNA through ROS. Ionizing radiation activates the DNA damage response, resulting in DNA repair, cell cycle arrest, and different forms of cell death. Abbreviations: AKT1, AKT serine/threonine kinase; ATM, ataxia-telangiectasia mutated; ATR, ataxia-telangiectasia and Rad3-related; CDK, cyclin-dependent kinase; CHK, checkpoint kinase; DNA-PK, DNA-dependent protein kinase; GPCR, G-protein-coupled receptor; MRN complex, Mre11-RAD50-Nbs1 complex; mTOR, mammalian target of rapamycin; PARP, poly (ADP-ribose)-polymerase; PI3K, phosphatidylinositol 3-kinase; PTEN, phosphatase and tensin homolog; ROS, reactive oxygen species; RTK, receptor tyrosine kinase; WEE1, Wee1-like protein kinase.

**Figure 2 cancers-15-03585-f002:**
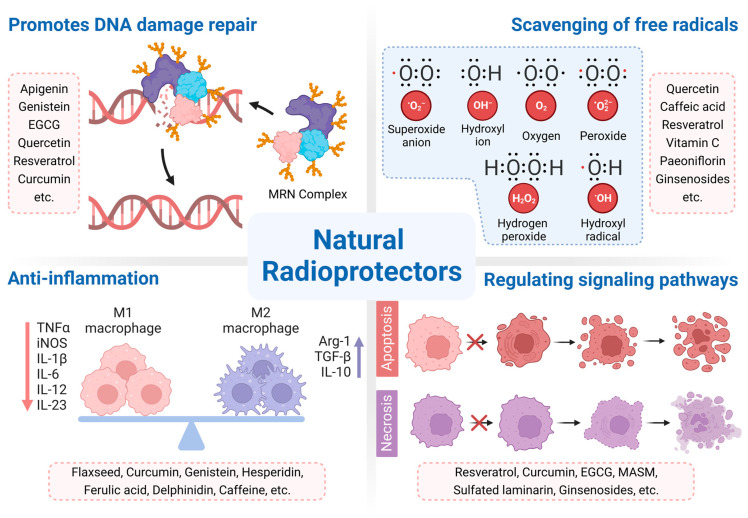
Summary of the radioprotective mechanisms of natural products. Abbreviations: Arg-1, arginase-1; EGCG, epigallocatechin gallate; IL, interleukin; iNOS, nitric oxide synthase; MASM, (6aS, 10S, 11aR, 11bR, 11cS)-10-methylamino-dodecahydro-3a, 7a-diazabenzo (de) anthracene-8-thione; MRN complex, Mre11-RAD50-Nbs1 complex; TGF-β, transforming growth factor-β; TNF-α, tumor necrosis factor-α.

**Figure 3 cancers-15-03585-f003:**
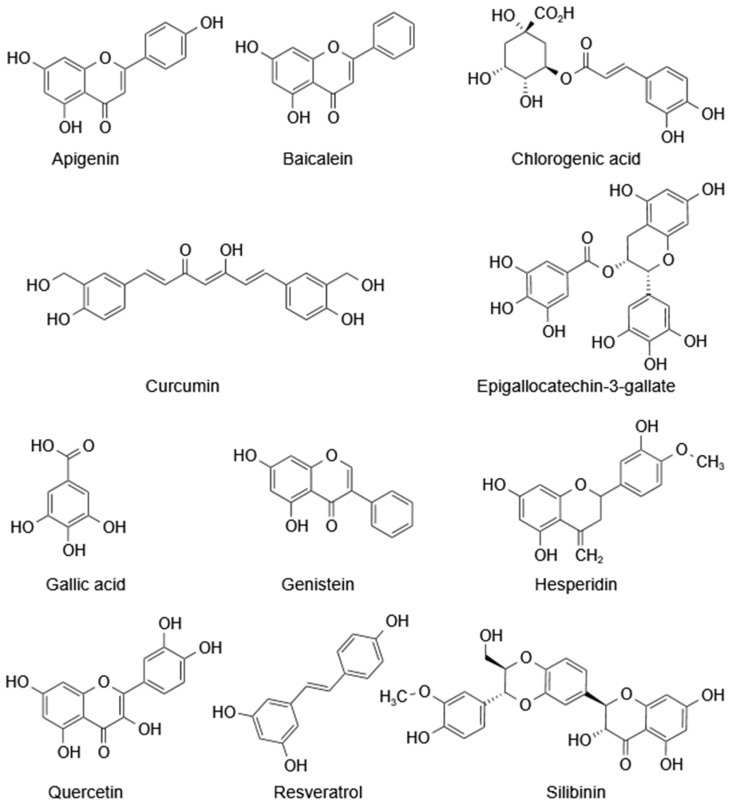
The chemical structures of polyphenols with radioprotective effects.

**Table 1 cancers-15-03585-t001:** Natural radioprotective agents and their mechanism of action.

Group	Agent	Natural Sources	Mechanism of Action	Trial Status	Reference
Polyphenols	Genistein	Soybean	Free radical scavenging; anti-inflammatory	I, II	[119,215]
	EGCG	Tea, Camellia sinensis	Free radical scavenging; inhibiting lipid peroxidation, ROS generation and apoptosis; reducing DNA damage; regulating immune activity	I, II	[127,128,129,130,131,216]
	Quercetin	Berries, shallots	Free radical scavenging; increasing DNA damage repair	II	[25,136,137]
	Resveratrol	Soy, grapes, peanuts	Free radical scavenging; inhibiting lipid peroxidation and apoptosis; regulating antioxidant enzymes	I, II, III	[27,28,90,217,218]
	Curcumin	Curcuma longa	Free radical scavenging; inhibiting lipid peroxidation; regulating antioxidant enzymes; anti-inflammatory	I, II, III	[92,178,219]
Polysaccharides	ASP	Astragalus	Reducing ALT, AST, and LDH levels	II	[32,33]
	SP	Schisandra	Protecting immunoglobulins and inhibiting apoptosis	Preclinical	[34]
	HSP	Hohenbuehelia serotina	Blocking the mitochondrial apoptosis pathway	Preclinical	[35,191]
Alkaloids	Matrine	Sophora flavescens	Modulating radiation-induced multiple signaling pathways	Preclinical	[38]
	Ligustrazine	Rhizoma Chuanxiong	Inhibiting ROS generation and apoptosis; reducing DNA damage	Preclinical	[39]
	β-Carboline	Picrasma quassioides	Free radical scavenging	Preclinical	[201]
Saponins	Ginsenosides	Ginseng	Inhibiting apoptosis	Preclinical	[36]
	ASS	Acanthopanax senticosus	Free radical scavenging; regulating immune activity	Preclinical	[37]
Others	Sesamol	Sesame, sunflower	DNA damage repair; inhibiting lipid peroxidation	Preclinical	[209,210,211]
	Vitamin C	Citrus fruits, berries	Antioxidant; free-radical scavenging	I, II, III	[220,221,222]
	Vitamin E	Vegetable oils, nuts	Antioxidant; free-radical scavenging, inhibiting apoptosis	I, II, III	[220,223]

Abbreviations: ALT, alanine aminotransferase; ASP, Astragalus polysaccharides; ASS, Acanthopanax senticosus saponins; AST, aspartate aminotransferase; EGCG, epigallocatechin gallate; HSP, Hohenbuehelia serotina polysaccharide; LDH, lactate dehydrogenase; ROS, reactive oxygen species; SP, Schisandra polysaccharides.

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
