# Peer review of "Exploring Natural Products as Radioprotective Agents for Cancer Therapy: Mechanisms, Challenges, and Opportunities"

_cancers, 2023, doi:10.3390/cancers15143585_

Round 1

Reviewer 1 Report

1.  The authors must merge the summary in the abstract and delete the repetition and correct the sentence in line-19. “In addition, we discussed the challenges and opportunities for the use of natural products as novel radioprotective agents”.

2.  Update the abstract by inserting information about the radioprotective agents which are discussed in the manuscript.

3.  Write the reference for “Modern radiotherapy techniques, such as three-dimensional conformal radiotherapy (3D-CRT), intensity-modulated radiotherapy (IMRT), and stereotactic body radiation therapy (SBRT), help to maximize lethal effects on tumors while minimizing toxic effects on surrounding normal tissue”.

4.  Insert the citation- https://www.mdpi.com/2073-4409/11/14/2209 for interest has developed in investigating natural products, which are extracted from medicinal plants and are less toxic and less expensive, as potential radioprotective agents”.

5.  Write the reference https://onlinelibrary.wiley.com/doi/10.1002/ptr.7216 in the end of line-96 .

6.  Insert more information about natural products into the introduction.

7.  The whole manuscript lacks focus as per the tile. The authors wrote more general information in Section 2.1. Promoting DNA damage repair rather than natural products. This section needs to add more about the natural products for promoting DNA damage repair and delete the general information.

8.  Figure 2. should be redrawn by BioRender by inserting the name of some specific radioprotective natural agents into the figure for easy reading.

9.  Reference 51 is too old replace it by inserting, https://www.mdpi.com/1422- 0067/22/22/12455.

10.  The authors must update table1 with the addition of a reference column and cite the reference for each listed natural radioprotective agent in the table. For example, EGCG doi: 10.3389/fnut.2022.1078642. PMID: 36712528; PMCID: PMC9874859.

11.  The entire manuscript should be updated with more recent references.

12.  Remove inconsistent spacing in whole draft.

13.  A major English language correction is required.

Please check the plagiarism and self citations inserted by authors

Author Response

Dear Esteemed Reviewer,

On behalf of all the contributing authors, I would like to extend our heartfelt gratitude for the incisive and constructive feedback you provided on our manuscript, titled “Exploring Natural Products as Radioprotective Agents for Cancer Therapy: Mechanisms, Challenges, and Opportunities” (Manuscript ID: cancers-2446325). Your insights and expertise have been instrumental in guiding revisions that have substantially improved the clarity and rigor of our work.

We greatly appreciate the time and effort you have invested in reviewing our work, and we would like to assure you that we have taken each of your comments to heart. Your observations regarding the abstract, references, and language corrections, among other aspects, have been addressed earnestly. For ease of reference, all the amendments have been highlighted in blue in the attached revised manuscript. Alongside, please also find enclosed a Point-By-Point Response that outlines the changes made in response to each of your comments.

I would like to assure you that we have taken all the points into consideration and have made substantial revisions to the manuscript in accordance with your recommendations. We believe that these modifications significantly improve the coherence, rigor, and contribution of our work.

Once again, thank you for your invaluable input. Your dedication to maintaining high standards of academic integrity and scholarship is commendable and greatly appreciated.

Should you have any further questions or require additional clarifications, please do not hesitate to contact me.

Warm regards,

Bingwen Zou M.D.

Jun 29th, 2023

Reviewer 2 Report

An interesting manuscript for radioprotectives, which probably could be a new type of drug. However, several drawbacks appear in this review.

I miss a paragraph discussing previous reviews in the field.

 I also miss an initiating statement of the number of such approved drugs.

Some compounds are praised for their radioprotective effects. However, no statement of the tested concentrations is given. I looked up two such compounds and observed that they were tested in concentrations of several hundred µg/ml making the observation of no clinical relevance. This observation is missing from the review.

I miss references to previous reviews in this field and why this review is needed.

Line 17 and 29: Natural products with low cost. All drugs must undergo severe and expensive assays and clinical trials meaning that no “low cost” drug will be ever developed. In addition, I would be surprised if any pharmaceutical company would develop a drug based on an observation that is not patented. Thus, I can not accept a statement like natural products are cheap. Firstly, isolation and standardization by itself will have certain costs and then secondly, the expenses for performing the clinical trials I to III must pay off. In addition, are expenses for intellectual rights. I miss a statement if any natural products have passed the clinical assays.

Below do also find some examples of bad language.

The Abstract and the Summary are very similar.

Line 41 modality

Line 65 can cause toxicity in adjacent tissues, might damage adjacent tissues.

Line 79: where does the legend to fig. 1 end and where does the txt start?

Line 99 owing to their molecular structure. Of course, any drug has its effect because of its structure.  This statement offers no new insight.

Line 117, I miss a reference.

Created with BioRender. No advertisements in Scientific literature. Of cause the authors have created these figures, otherwise, it would violate copyrights.

Line 145. Ionizing radiation can cause the generation of ROS from water molecules, including •OH, H2O2, and •NO Creation of NO would mean the formation of nitrogen from either hydrogen of oxygen. Both possibilities are impossible.

Line 188 can be subgrouped into flavonoids and nonflavonoids (Figure 3). Such a statement is trivial. Of cause this is the case.

Line 232: this is important information.

Line 232 Genistein also possesses estrogenic activity.

Line 317 Why are anthocyanins listed as nonflavonoids?

Line 348 Plant polysaccharides are widely found in plant organs, it would be surprising if they were found in animal organs.

Line 349: immunomodulatory: do plants have an immune system?

Line 363: Some natural products are praised for biological activities. However, it should also be included in which concentrations these effects are found. Very often exceedingly high concentrations are reported excluding any clinical use. In the case of astragalus polysaccharide concentrations of several 100 µg/ml make the findings trivial and not clinically relevant. This also is true for laminaran polysaccharides. Similar other concentrations should be mentioned by the authors.

See other comments.

Author Response

Dear Esteemed Reviewer,

We deeply appreciate your comprehensive review and invaluable feedback on our manuscript, titled "Exploring Natural Products as Radioprotective Agents for Cancer Therapy: Mechanisms, Challenges, and Opportunities" (Manuscript ID: cancers-2446325). Your comments have served as a significant catalyst for enhancing the quality of our work, and we are grateful for your expert insights.

Following your feedback, we have performed thorough revisions to our manuscript, ensuring that every point you raised has been adequately addressed. In particular, we have made improvements to the abstract, fortified our reference list, and engaged a professional language editing service to provide a comprehensive review of the manuscript.

We have also provided a Point-By-Point Response for a detailed overview of the revisions made in response to each of your comments. To facilitate easy reference, these modifications have been highlighted in blue in the attached revised manuscript.

We firmly believe that the revisions undertaken, guided by your astute recommendations, have significantly bolstered the coherence, rigor, and contribution of our manuscript.

Thank you once again for your diligent review and meaningful contribution towards the improvement of our manuscript. We deeply value your dedication to promoting excellence in scholarly research.

Please do not hesitate to contact me if you require further clarifications or additional information.

Warm regards,

Bingwen Zou, M.D.

Jun 29th, 2023

Round 2

Reviewer 1 Report

The authors have significantly improved the manuscript. Before the publication, there are still some flaws that should be addressed.

  1. As suggested to the authors earlier, the heading simple summary and contents still need to be removed. There should be only an abstract covering all the information.
  2. A major English language correction is required.

A major English language correction is required.

Author Response

Dear Esteemed Reviewer,

On behalf of all the contributing authors, I would like to express our gratitude for your continued support and guidance in reviewing our manuscript, titled "Exploring Natural Products as Radioprotective Agents for Cancer Therapy: Mechanisms, Challenges, and Opportunities" (Manuscript ID: cancers-2446325). We are pleased to know that you have acknowledged the significant improvements in the manuscript. Your feedback has been invaluable in refining the quality and focus of our work.

With respect to the suggestion to eliminate the "Simple Summary" heading and its content, we understand the intent behind this recommendation and have considered it carefully. However, we were unable to remove the summary in this revision for specific reasons, which will be explained in detail at a later stage. As for the language, we understand that further revisions are necessary to improve the clarity and readability of our manuscript. We have sought professional English language editing services to ensure that the text is grammatically correct and communicates our research clearly to readers. We believe that this major language correction will significantly improve the manuscript. In addition, to distinguish this round of revisions from the first, all the current revisions have been marked in green.

We are committed to ensuring that our manuscript meets the highest standards and appreciate your constructive feedback throughout this process. We hope that the further revisions we have made address your concerns.

If you would like more information or have any questions, please do not hesitate to contact me.

Warm regards,

Bingwen Zou M.D.

Jul 4th, 2023

Reviewer 2 Report

The authors have answered some of my questions. I appreciate the text Line 627 to 642, however, I asked for the concentrations of natural products needed for a radiation-limiting effect. Here the authors have answered my question concerning the two examples I mentioned but not all the other natural products. I suggest the authors look up all the examples they mention and check that the needed concentrations a relevant for clinical use.

In addition, I am not eligible to check all biological parts of the manuscript. It appears that I am the only reviewer. I suggest an expert with biological expertise also review the manuscript.

Some minor comments:

Line 188: Not all natural products can be used. Maybe the word some should be included.

Line 238 Innula racemosa in italic

No comments.

Author Response

Dear Esteemed Reviewer,

We are deeply appreciative of the time and effort you have invested in providing valuable feedback on our manuscript. Your comments have indeed helped us identify areas where further detail and clarity are required.

With regard to your observation concerning the concentrations of natural products necessary for a radiation-limiting effect, we acknowledge the oversight in our initial response. In line with your suggestion, we have reviewed the natural products mentioned in the manuscript to provide comprehensive information about the relevant concentrations needed for clinical application. We believe this will significantly enhance the scientific rigour and practical relevance of our study.

As for your minor comments, we are grateful for your attention to detail. Specifically, in line 188, we have included the word 'some' to accurately reflect that not all natural products can be used for radioprotection. Moreover, as per your suggestion, we have now italicised "Innula racemosa" on Line 238. To distinguish this round of revisions from the first, all the current revisions have been marked in green.

Your insightful comments are invaluable to the improvement of our manuscript. We are dedicated to making the necessary changes and are confident that these will further improve the quality and impact of our work.

Once again, we extend our heartfelt appreciation for your time and critical assessment of our manuscript. We look forward to any further recommendations or queries you may have.

Sincerely,

Bingwen Zou, M.D.

Jul 4th, 2023
